# Variable Gap Sealing Technology of a Hydraulic Cylinder Based on Magnetic Shape Memory Alloy

Xiaolan Chen [1], Fuquan Tu [2,*], Feng Gao [1], Heming Cheng [1] and Shixiong Xing [3]

1   School of Electromechanical and Automobile Engineering, Huanggang Normal University, Huanggang 438000, China; chenxiaolan@hgnu.edu.cn (X.C.); gfwh517@sohu.com.cn (F.G.); heming20148439@163.com (H.C.)
2   Key Laboratory of Metallurgical Equipment and Control Technology, Ministry of Education, Wuhan University of Science and Technology, Wuhan 430081, China
3   Hubei Zhongke Research Institute of Industrial Technology, Huanggang 438000, China; xingshixiong_123@163.com
*   Correspondence: tufuquan@wust.edu.cn

**Abstract:** The synergistic control of resistance reduction and sealing poses challenges to enhancing the rapid dynamic response ability of servo hydraulic cylinders; the key to solving this problem is effectively controlling the sealing gap value. In this study, a micro-variation between the hydraulic cylinder and the piston based on the disadvantage of conventional seals, constant gap seals, and lip gap seals was constructed; MSMA assist support blocks were designed on the piston to form a gap seal strip; then, the sealing gap value could be changed by controlling the magnetic field intensity. Simultaneously, the effects of magnetic field strength, parts-manufacturing precision, temperature, and hysteresis on the micro-variation in the MSMA were analyzed, and effective solutions were proposed. Finally, experiments on the magnetic field, temperature, and hysteresis were conducted by the measurement system. The results showed that the variable value of the sealing gap with the MSMA is feasible under ideal conditions, and can effectively change the amount of MSMA expansion by controlling the magnetic field strength, temperature, preload, etc., and then change the amount of the sealing gap of the hydraulic cylinder. This is the key to achieving friction and sealing control, which plays a crucial and active role in improving the efficiency of hydraulic systems. However, the impact of hysteresis effects cannot be ignored, which will be the main problem to be solved in the future.

**Keywords:** hydraulic cylinder; gap sealing; MSMA; electromagnetic drive effect





## 1. Introduction

Hydraulic transmission uses liquid as the working medium to transmit motion, and has become one of the forms of transmission in the field of mechanical engineering. These systems are composed of a power unit (hydraulic pumps), actuators (hydraulic cylinders), control elements (hydraulic valves), auxiliary components (tubes), and a working medium (hydraulic oil). Hydraulic transmission provides large output power per unit mass and high power transmission. However, the low efficiency of the hydraulic system is caused by internal and external leakage. The operating efficiency of actuators is determined by their response speed and performance, which are responsible for the external work and convert hydraulic energy into mechanical energy [1]. Therefore, their working efficiency must be improved, becoming a hot issue for many researchers.

The frequency response of the hydraulic cylinder is related to its sealing forms, of which there are three main types. First, the traditional hydraulic cylinder is sealed with an elastic ring, as shown in Figure 1a, which can improve the volumetric efficiency, but it considerably increases the frictional resistance. Second, the gap sealing method is applied, as shown in Figure 1b, which is formed by the flow resistance, the tension between the oil

and the mating surface, the molecular force, etc. However, the constant gap seal increases the amount of leakage. Third, now, the lip structure is designed at both ends of the piston, as shown in Figure 1c, which opens outward with increasing pressure difference between the two ends of the piston, thereby reducing the sealing gap and the leakage [2].

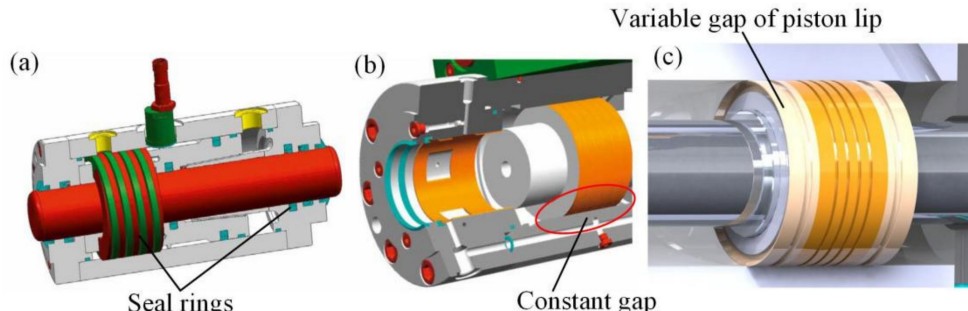

**Figure 1.** Three sealing forms for hydraulic cylinders: (**a**) the sealing ring; (**b**) the constant gap seal; (**c**) the variable gap seal.

Unfortunately, when the pressure difference is small, the amount of leakage is large. Conversely, if the pressure difference is too large, the piston lip opens outward too much, causing the piston to contact the inner surface of the cylinder and result in lip deformation failure, as shown in Figure 2.

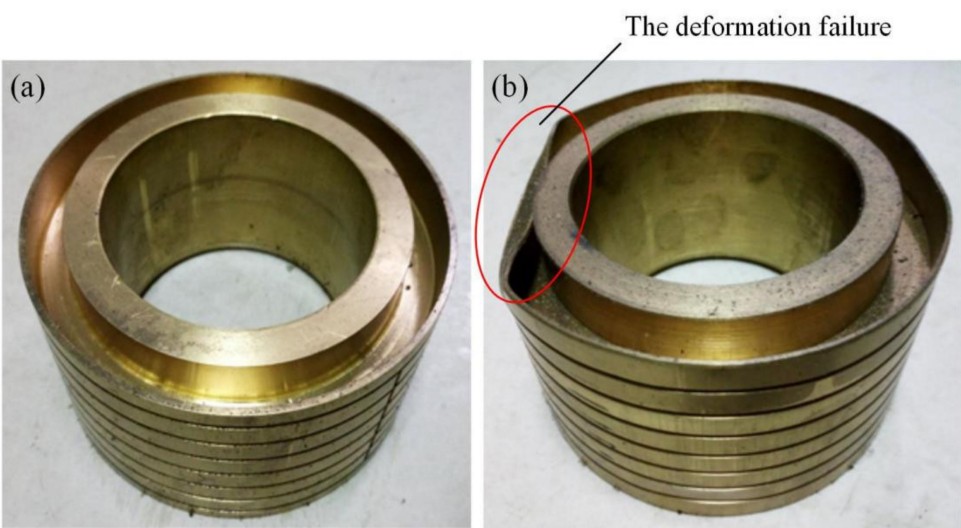

**Figure 2.** The structure of the piston lip: (**a**) normal and (**b**) deformed piston lip structure.

Chen and colleagues [3,4] proposed bionic micro-texture processing on the inner surface of hydraulic cylinders using dynamic texture lubrication to improve the above conditions, and analyzed the wedge dynamic pressure effect through simulation. However, it is difficult to process the bionic micro-texture on the annular inner surface of a cylinder block, so this method has not yet been experimentally applied.

In recent years, more researchers have been focusing on using new functional materials to improve the response speed of hydraulic components. Functional materials refer to materials that have specific functions through the action of light, electricity, magnetism, heat, chemistry, biochemistry, etc.

Magnetic shape memory alloy (MSMA) is a new and promising type of functional material, and has attracted considerable attention from researchers because of its large recoverable strain (micro deformation), fast response speed, and high-cycle fatigue properties [5,6]. As a kind of magnetic functional material, MSMA has been widely used in

the hydraulic industry in recent years, as the key component of actuators [7,8], sensors [9], energy harvesters [10,11], fluid control [12], hydraulic micropumps [13,14], and some other devices [15].

However, the expansion effect of MSMA has never been applied to the micro clearance compensation of hydraulic components. Therefore, we propose using the electromagnetic drive expansion effect of MSMA to control the sealing gap of the hydraulic cylinder and then to achieve the purpose of reducing internal leakage.

## 2. Electromagnetic Drive Mechanism of MSMA

As we know, the MSMA is a kind of electromagnetic drive material and exhibits a shape-memory effect that can be strained in an external magnetic field, including Fe-C, Fe-Pd, Fe-Mn-Si, Co-Mn, Fe-Co-Ni-Ti, and intermetallics based on the Heusler phase $Ni_2MnGa$ [16]. The MSMA essentially uses a magnetic field to control the reorientation behavior of the twins [17]. When it is under the action of a magnetic field, the twin boundary will move, and the twin crystal variants will be rearranged, causing a large deformation in Figure 3.

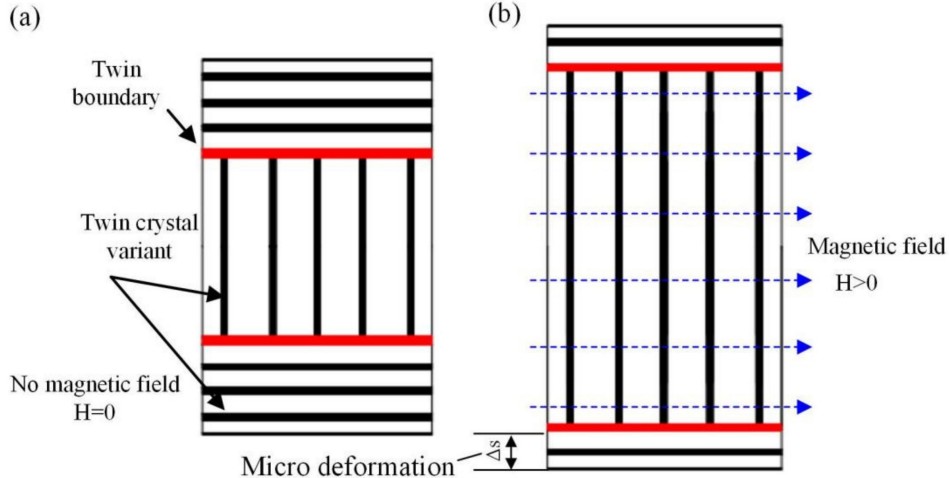

**Figure 3.** Deformation principles of MSMA: (**a**) the original shape of the MSMA without a magnetic field; (**b**) the strained shape with a magnetic field.

The term $H$ in Figure 3 is the magnetic field strength, where each tile represents a twinned crystal associated with the vector of the magnetic moment. When the applied magnetic field is zero, the magnetic moment of the twin crystal variants is not aligned in the same direction. When the horizontal magnetic field applied to the MSMA reaches a certain value, the magnetic moment associated with each twin crystal variant produces a rotation to match the direction of the external magnetic field. An increase (defined as $\Delta s$) in the length of the variant perpendicular to the direction of the magnetic field causes deformation of the entire crystal block, thereby achieving a macroscopically visible amount of deformation.

The MSMA is a new type of smart material that has both a magnetic shape memory effect and a thermoplastic shape memory effect. Compared to giant magnetostrictive materials (GMMs) [7] and piezoelectric ceramic materials (PCMs) [18], it has higher response frequency and a higher deformation rate. Taking $Ni_2MnGa$ as an example, its material properties are shown in Table 1.

**Table 1.** Properties of Ni$_2$MnGa.

| Properties of Material | Values |
| --- | --- |
| elongation in magnetic field (percentage) | 3–6% |
| response frequency (Hz) | 1000–2000 |
| force density (MPa) | 2 |
| material density (g/cm$^3$) | 8 |
| temperature coefficient (K) | 0.003 |
| Martensite transformation temperature (°C) | 50 |
| cycle fatigue (times) | 1,000,000,000 |
| Curie temperature (°C) | 95–105 |

## 3. The Hydraulic Cylinder Clearance Compensation Based on MSMA

### 3.1. The Structure Design of Gap Compensation

Friction and leakage cause problems in the normal operation of hydraulic cylinders, which are affected by the clearance between the piston and cylinder. Small gaps lead to large frictional resistance, whereas large gaps cause serious leaks. The amount of leakage also increases with increasing pressure differences. Therefore, effective control of the seal gap is the key to solving this problem.

The clearance compensation idea of the MSMA auxiliary support block was designed in Figure 4. A plurality of annular grooves is designed on the outer surface thereof according to the length of the piston, and several MSMA auxiliary support blocks are continuously arranged in the annular groove, thereby constructing a few gap seal rings.

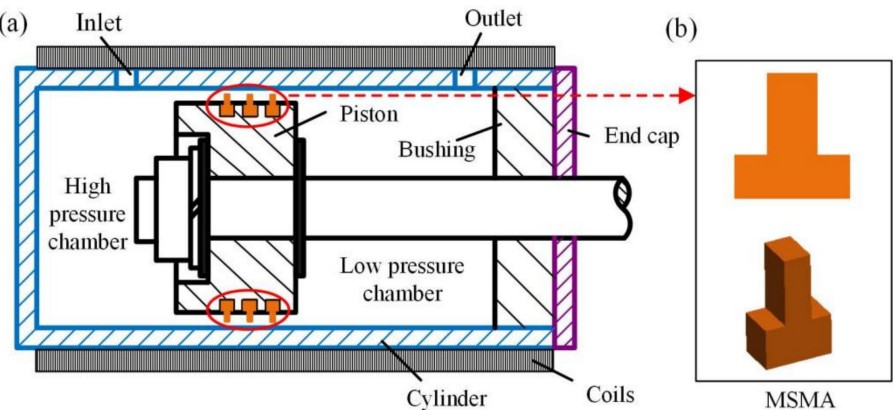

**Figure 4.** Clearance compensation of the hydraulic cylinder based on MSMA: (**a**) the structure design of gap compensation; (**b**) a single MSMA auxiliary support block.

### 3.1.1. The Structure Design of the MSMA Auxiliary Support Block

We designed the MSMA auxiliary support block in a convex shape, and they are continuously arranged in the outer annular groove of the piston, as shown in Figure 4b. The outermost side of the annular groove is fixed by an open piston ring matched with the convex MSMA auxiliary support block. When an electromagnetic field is applied near the piston, the convex blocks are extended outward, compensating for the gap between the cylinder and the piston, solving the problem of the increase in the leakage amount when the hydraulic cylinder gap is sealed.

To ensure the uniformity and control of the variable clearance, a strict requirement exists on the concentricity of the hydraulic cylinder and piston. The MSMA auxiliary support block must also have a certain manufacturing accuracy and positioning accuracy, e.g., the diameters of the cylinder and piston are 40 mm ($\phi40H6(^{0}_{+0.016})/f5(^{-0.036}_{-0.025})$).

### 3.1.2. The Principle of the Magnetic Circuit

Since the clearance compensation direction of the MSMA seal rings is radial, and the telescopic direction is perpendicular to the direction of the magnetic field, it was necessary to design a coil structure capable of generating a magnetic field parallel to the axis of the hydraulic cylinder. Thus, the toroidal coils were designed on the outer surface of the cylinder. Considering the stability of the magnetic field strength, the toroidal coils in Figure 4a are elongated in the axial direction to cover the entire piston movement stroke.

When the coil is energized, a magnetic field parallel to the axis is created inside its annulus. Changing the magnitude of the energization current results in different-strength magnetic fields.

According to Faraday's electromagnetic law, the input voltage ($V_m$) of the excitation coil can be expressed as [19]:

$$V_m = Ri + NL_d\frac{di}{dt} + N\frac{d\Phi_r}{dt} \tag{1}$$

where $R$ is the coil resistance, $i$ is the coil current, $N$ is the number of coil turns, $L_d$ is the coil inductance, and $\Phi_r$ is the magnetic flux through the MSMA.

Therefore, by changing the voltage and then controlling the change in the current, the change in the magnetic field strength is achieved.

### 3.2. Governing Equations

By analyzing the working principle of the MSMA support block and the work characteristics, the system can be simplified to the equivalent mechanical model shown in Figure 5, where $m$ is the total weight of the MSMA support block and hydraulic oil, $F$ and $x$ are the output force and displacement of MSMA, respectively; $K$ is the equivalent spring stiffness of the recovered magnetic field; and $C$ represents the viscous resistance of the hydraulic oil.

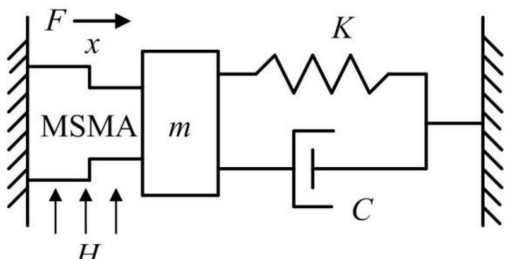

**Figure 5.** Equivalent mechanical model of the system.

The constitutive model of MSMA describes the process of material strain in a magnetic field, including elastic constitutive models and twin boundary slip models. Since the elastic constitutive model is within the linear strain range of the material, its expression is [20]:

$$[\varepsilon] = [e][\sigma] \tag{2}$$

where the symbol $[\varepsilon]$ is the strain of the material, $[e]$ is the elastic compliance coefficient of the material, and $[\sigma]$ is the stress applied to the material. Thus, the stress ($\sigma_{\text{mag}}$) to which the material is subjected in the magnetic field can be expressed as:

$$\sigma_{\text{mag}}(h) = \frac{\int_0^h m_t(h)dh - \int_0^h m_a(h)dh}{\varepsilon_0} \tag{3}$$

where $h$ is the strength of the magnetic field in which the MSMA is located; and $\varepsilon_0$ is the material crystal strain limit; and $m_t(h)$ and $m_a(h)$ are the hard magnetization curve and

the easy magnetization curve of the material magnetization axis, respectively, which are given by:

$$m_t(h) = S_t h - S_t(h - h_t) \tag{4}$$

$$m_a(h) = S_a h - S_a(h - h_a) \tag{5}$$

where $S_t$ and $S_a$ are the slopes of the magnetization curves. The $h_t$ and $h_a$ are the saturation flux density, respectively, and can be defined as:

$$(h - h_t) = \begin{cases} h - h_t & h - h_t > 0 \\ 0 & h - h_t < 0 \end{cases} \tag{6}$$

$$(h - h_a) = \begin{cases} h - h_a & h - h_a > 0 \\ 0 & h - h_a < 0 \end{cases} \tag{7}$$

In actual work, the MSMA auxiliary support block is subjected to the oil pressure at the same time, which is defined as $\sigma_p$. Then, the actual stress of the MSMA is:

$$\sigma_t = \sigma_{\text{mag}}(h) - \sigma_p = \frac{\int_0^h m_t(h)dh - \int_0^h m_a(h)dh}{\varepsilon_0} - \sigma_p \tag{8}$$

Substituting Equations (4) and (5) into Equation (8), the magnetostatic stress equation of the material in the magnetic field can be expressed by:

$$\sigma_t(H, \sigma_p) = \frac{\int_0^H m_t(H)dH - \int_0^H m_a(H)dH}{\varepsilon_0} - \sigma_p = \frac{(S_a - S_t)H^2}{2\varepsilon_0} - \frac{S_a(H - H_a)^2}{2\varepsilon_0} + \frac{S_t(H - H_t)^2}{2\varepsilon_0} - \sigma_p \tag{9}$$

where $H_t = h_t/\mu_0$, $H_a = h_a/\mu_0$, and $\mu_0$ is the permeability of hydraulic oil with a value of $4\pi \times 10^{-7}$ H/m.

### 3.3. Key Factors Affecting the Expansion of MSMA

The first factor to be considered in micro clearance compensation using the magnetic drive deformation effect of MSMA is the magnetic field strength. The radial elongation and compensation of the MSMA auxiliary support blocks are directly determined on the magnitude and direction of the magnetic field strength. When the conductor of the electromagnetic mechanism is under the action of an alternating magnetic field, it will consume itself, mainly including hysteresis loss and eddy current loss.

The second factor affecting the expansion of MSMA is the structural and assembly tolerances of the hydraulic cylinder, which cause the hydraulic cylinder and the piston to be eccentric, thereby increasing the internal leakage [21].

Hydraulic oil overcomes the resistance flow and causes its temperature to rise, which is the third influencing factor. This heat is transferred to the MSMA auxiliary support block, which significantly impacts the amount of expansion.

The influences of these factors were analyzed, and some effective solutions are proposed in the subsequent sections.

### 3.3.1. The Magnetic Field Strength and Its Loss

Before the magnetic field strength reaches saturation, the higher the magnetic field strength, the greater the elongation of the MSMA, the smaller the gap between the hydraulic cylinder movement pairs, and the less the leakage. However, under the action of the alternating magnetic field, the relationship between the magnetic field strength $H$ and the magnetic induction intensity $B$ in ferromagnetic materials is mostly nonlinear or single-valued [22,23]. The change in magnetic induction intensity $B$ inside the material lags behind the change in magnetic field strength $H$, resulting in the hysteresis effect, as shown in Figure 6, increasing the leakage of the hydraulic cylinder during the initial movement. The research methods of the hysteresis effect mainly include the establishment of an

accurate hysteresis model and the use of empirical formulas. Many mathematical models are available, mainly from Preisach and Jiles-Atherton [24,25].

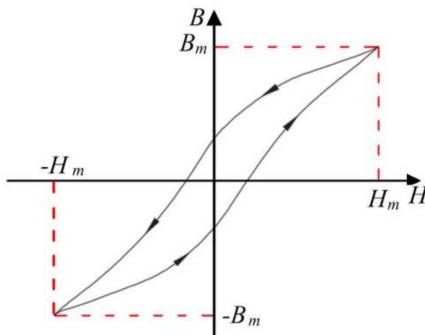

**Figure 6.** A diagram of the hysteresis effect.

Since the induced current forms a magnetic field, it hinders the change in the excitation magnetic field. Therefore, the excitation magnetic field needs to cancel the magnetic field generated by the eddy current to function, and the canceled energy is emitted in the form of heat, which is called eddy current loss. Zhu et al. found that minimizing the quadratic sum of the transmit-coil and receive-coil current is more effective in reducing the eddy current loss through simulation and experimental analysis [26]. The eddy current loss of the common iron core is also usually calculated by the empirical formula method, and it is transformed into the effect on the temperature rise of MSMA components. Therefore, when analyzing eddy current loss in actual work, it is usually converted into the effect of temperature.

### 3.3.2. Structural and Assembly Tolerances of the Hydraulic Cylinder

The initial clearance between the piston and the inner surface of the cylinder is 10–15 microns. Due to manufacturing errors, the clearance between the cylinder and the piston may increase. If the hydraulic cylinder block and the piston are eccentrically generated during assembly, the leakage amount will considerably increase, reaching 2.5 times the normal gap leakage [4]. Therefore, according to their team's bionic micro-texture dynamic pressure lubrication method, the eccentricity problem can be effectively improved, so the piston and the cylinder body are automatically centered under the action of dynamic pressure lifting. Considering the difficulty of bionic micro-texture processing on the inner surface of the cylinder, the micro-texture morphology necessary for generating the dynamic pressure lifting force can be processed on the MSMA auxiliary support ring, as shown in Figure 7.

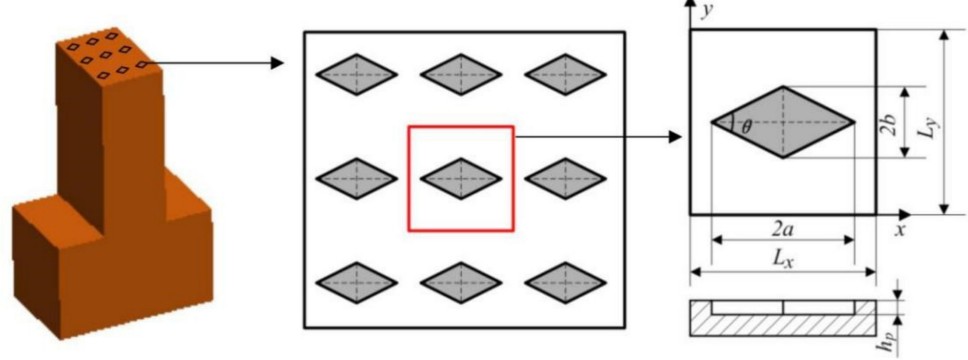

**Figure 7.** Micro textures on the surface of MSMA auxiliary support block.

### 3.3.3. Temperature Effect

MSMA has both a magnetic drive shape memory effect and a thermoplastic shape memory effect. Therefore, the temperature rise of the hydraulic oil has a non-negligible influence on its stretch. There are two main reasons for the increase in the temperature of hydraulic oil: the internal energy generated by the hydraulic oil against various resistances, and the heat generated by the eddy current loss of the exciting coil.

The internal energy against various resistances can be cooled by installing a cooling device (water tank or fan) at the tank position, which can effectively reduce the temperature of the circulating hydraulic oil. Simultaneously, a cooling device can be added to the excitation coil to offset the eddy current heat (Figure 8) [27].

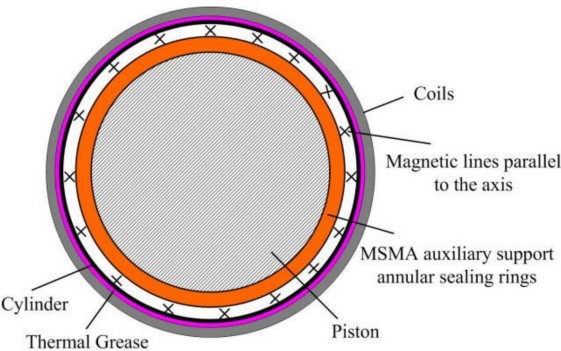

**Figure 8.** Schematic diagram of the excitation coil heat sink.

As shown in Figure 8, the thermal grease is filled in the gap between the cylinder block and the excitation coil to increase its heat dissipation performance. An auxiliary fan can also be set to cool.

The cooling device needs to be designed, but the actual temperature rise during calculation cannot be ignored. Therefore, it is necessary to introduce a temperature correction coefficient (defined as $k$) to Formula (9) [28].

$$\sigma_t(H, T, \sigma_p) = \frac{kT \cdot (S_a - S_t)H^2}{2\varepsilon_0} - \frac{S_a(H - H_a)^2}{2\varepsilon_0} + \frac{S_t(H - H_t)^2}{2\varepsilon_0} - \sigma_p \qquad (10)$$

where term $T$ is the ambient temperature.

## 4. Experiments and Discussions

$Ni_2MnGa$ was used as the sample for experiments with dimension of $5 \times 5 \times 15$ mm, which was purchased from the AdaptaMat Company in Finland (Figure 9).

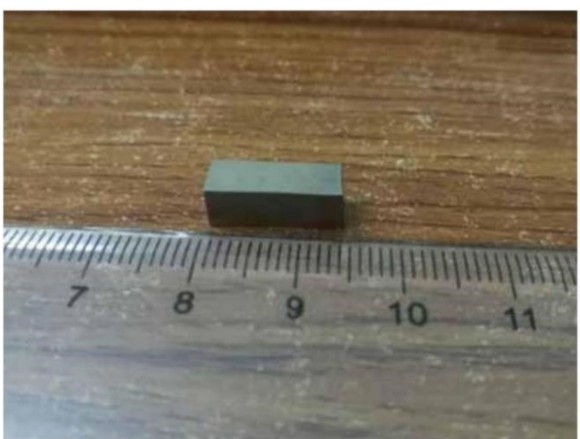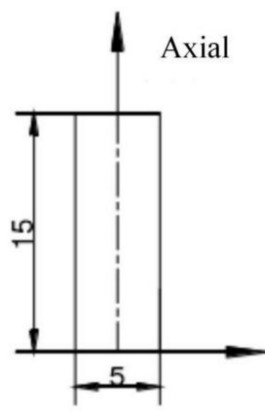

**Figure 9.** The sample used for experiments.

In the measurement system, the oil press on the MSMA was replaced by the sample preload. The sample preload was provided by the hydraulic system, which was measured by the force sensor with a load range of 0~2.5 MPa. The applied control magnetic field was provided by the energized current, and its magnetic field range as 0~1.5 Wb/m$^2$, which is proportional to the energized current. In addition, the operating temperature was adjusted by the oiled temperature loading system, and the operating temperature variation range was 16~35 °C, as shown in Figure 10.

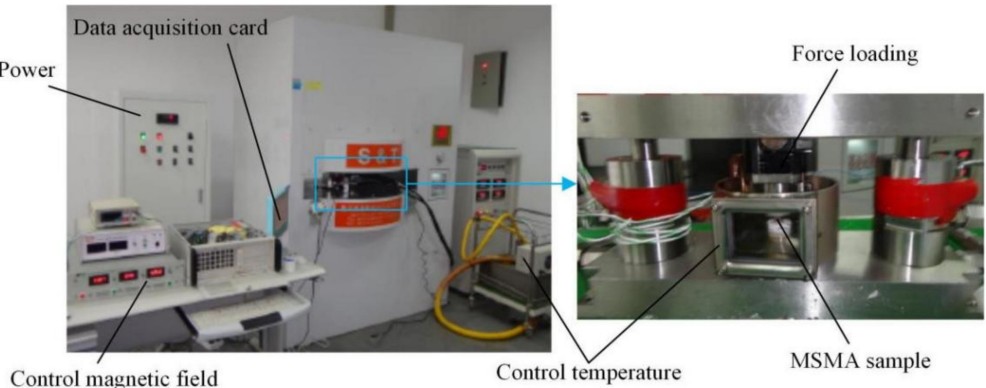

**Figure 10.** A diagram of the measurement system.

### 4.1. Effect of Magnetic Field on Deformation Rate

The working temperature and pre-stressing were kept constant during the experiment firstly, and then the magnetic field was controlled to change in the range of 0~1 Wb/m$^2$. Finally, the strain gauges were used to measure the strain rate of MSMA. The results are shown in Figure 11.

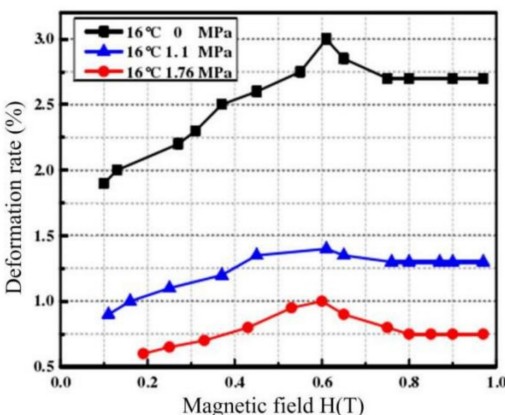

**Figure 11.** The relationship between deformation rate and magnetic field.

Figure 11 shows that under constant temperature and stress conditions, the MSMA elongation deformation rate and the magnetic field firstly increased linearly when it changed from 0.20 to 0.61 Wb/m$^2$. The MSMA elongation deformation rate decreased slowly first from 0.61 to 1 Wb/m$^2$ and then hardly changed.

The magnetic saturation phenomenon of MSMA can be explained by the magnetic drive mechanism. The direction of the Ni$_2$MnGa crystal structure is reselected under the magnetic field. The short axis of the twins indicates the direction that is easy to be magnetized. When the magnetic field is equal to zero, the twin axis follows the long axis of the twins. Then, the magnetic field gradually increases and the other twin variants grow at the same time. When the magnetic field is larger, the crystal is easily rotated by the magnetized axis to the direction of the external magnetic field.

### 4.2. Effect of Temperature on Deformation Rate

Firstly, the magnetic field and pre-stressing were kept constant during the experiment, and then the operating temperature was changed within the range of 16–35 °C. Finally, strain gauges were used to measure the deformation rate of MSMA. The results are shown in Figure 12.

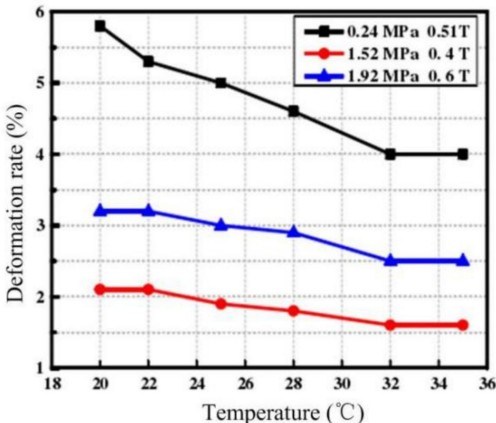

**Figure 12.** The relationship between deformation rate and temperature.

We found that the deformation rate of MSMA decreases gradually with increasing temperature and then remains unchanged, because the deformation rate of MSMA is affected by the temperature due to the transformation of martensite. The MSMA effect can only occur in the low temperature phase (martensitic phase). Therefore, the temperature rises to the high temperature phase (austenite phase) and the magnetic strain effect disappears. When the temperature rose from 16 to 32 °C, the microstructure in the MSMA alloy begins to change from martensite to austenite. During the transformation, all the alloy inside is a mixture of martensite and austenite. The MSMA shrinkage deformation rate gradually decreases. When the temperature rises above 32 °C, the MSMA alloy has an austenite structure. At this time, MSMA has no magnetically induced strain effect, so the shrinkage deformation rate of MSMA remains unchanged.

### 4.3. Hysteresis Effect

The working temperature and pre-stressing were kept constant during the experiment at first, and then the magnetic field was controlled to change in the range of 0~1.5 Wb/m$^2$. Finally, the eight strain gauges arranged in the whole bridge were used to measure the deformation rate of MSMA. The results are shown in Figure 13.

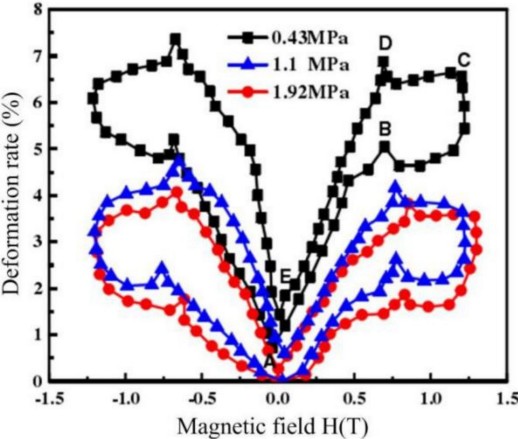

**Figure 13.** Hysteresis effect.

The magnetic field increased from zero, and the deformation rate of MSMA rose along the AB segment. As the magnetic field continued to increase, the deformation rate slowly increased along the BC segment. After reaching point C, the deformation rate was the largest. Starting from the magnetization saturation point C, the magnetic field decreased, and the deformation rate returned directly to point E along with CD and DA.

The reason why MSMA has the above hysteresis characteristics is that the magnetization and demagnetization processes are under the control of the magnetic field. In general, hysteresis occurs because the magnetic induction intensity is delayed compared to the change in the magnetic field during the demagnetization process. That is, the hysteresis characteristic of MSMA occurs during the demagnetization of MSMA.

## 5. Conclusions

Elongation of MSMA is possible when an external magnetic field is applied, and many studies and experiments confirmed the feasibility of micro-deformation by MSMA in driving hydraulic actuators. Therefore, further exploration of the application of the new functional material MSMA in the field of hydraulics is the future research to which we are committed.

In this paper, the idea of using MSMA to produce micro-deformation between the hydraulic cylinder and the piston was innovatively proposed based on the shortcomings of traditional seals, constant gap seals, and lip gap seals. The following conclusions were drawn from our design analysis and experimental results:

i.  Variable gap seal structures of the hydraulic cylinder, including the structure of the auxiliary support blocks, the mounting method, and the magnetic field generation form, were designed based on the specific working principle of the hydraulic cylinder, which helped to improve leakage in the hydraulic cylinder.

ii.  The factors influencing the micro-deformation of MSMA were analyzed and effective solutions and approaches were proposed.

iii.  The influence of the magnetic field and temperature in MSMA experiments showed the feasibility of using MSMA to realize the variable gap sealing function of hydraulic cylinders. Leak suppression in hydraulic cylinders can be achieved by controlling the magnetic field and temperature.

MSMA has an important, positive effect on enhancing the clearance compensation and improving the working efficiency of the hydraulic system. This provides theoretical support for the variable micro-gap compensation of hydraulic cylinders and has important practical application value.

We must also acknowledge that the work pressure and micro-dynamic pressure effect of bionic texture, especially their coupling effects, will also affect the MSMA variable gap seal. This is also another problem that needs to be solved in the future.

**Author Contributions:** Software, F.G. and H.C.; formal analysis, F.T.; investigation, F.T.; data curation, H.C.; writing—original draft preparation, X.C.; writing—review and editing, S.X.; funding acquisition, X.C. All authors have read and agreed to the published version of the manuscript.

**Funding:** The authors are grateful for financial support from Key Laboratory Open Fund (Study on the Mechanism of Gap Sealing Hydraulic Cylinder Based on Intelligent Materials No.MECOF2021B05), Central Guidelines Local Science and Technology Development Project of Hubei (Modular Refactoring Intelligent Manufacturing Key Technology and Verification Platform R & D and Application, No.2020ZYYD018) and our school-level research project (Research on Coupling Bionic Drag Reduction of Magnetic Control Deformation Suppression Mechanism of High Speed Servo Hydraulic Cylinders, No.202001103).

**Institutional Review Board Statement:** Not applicable.

**Informed Consent Statement:** Not applicable.

**Data Availability Statement:** The data presented in this study are available on request from the corresponding author. The data are not publicly available as the data also form part of an ongoing study.

**Acknowledgments:** Thanks to Wuhan University of Science and Technology to provide experimental equipment and instrument assistance in the experiment. Thanks to Huanggang Normal University to support in software simulation. Thanks to Renbo Xu to provide experimental analysis and guidance.

**Conflicts of Interest:** The authors declare no conflict of interest.

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
