# Peer review of "Variable Gap Sealing Technology of a Hydraulic Cylinder Based on Magnetic Shape Memory Alloy"

_coatings, doi:10.3390/coatings11080950_

Round 1

Reviewer 1 Report

Dear Authors,

you will find my comments below:

- lines 17, 75, 78,  - MSM alloys are not magnetostrictive materials - the deformation mechanism of the Ni2MnGa in the external magnetic field is completely different when compared to magnetostriction
'the way to get a large MFIS is to develop (1) the magnetically ordered SMA materials exhibiting a highly mobile twin microstructure, where the true magnetostriction (either spontaneous or forced) in the mechanically soft martensitic phase triggers the twins reorientation and/or (2) the magnetically ordered materials showing the field-induced reversible structural transformations triggering the volumetric strain or the shape recovery. When considering case (1), one should admit that a conventional MS effect is fully elastic, where the strain is directly proportional to the squared magnetic field (analog of a Young´s modulus equation), whereas the term MFIS implies a sum of the MS strain (which in practice may be neglected) and the incomparably large strain resulting from detwinning.' (https://doi.org/10.1016/B978-0-12-815732-9.00050-4)
- Figure 1 is blurry, propper version should be inserted
- line 91 - the applied magnetic field is horizontal in the Fig.3, not vertical
- line 134 - I suppose the words 'magnetic field' are missing in the phrase
- lines 144, 145, 151, 152, 153, 166, 172, 183, 204, 205, 206, 207, 258 - the symbols should be written in italic to distinguish them from the ordinary text
- lines 186, 239 - 'the magnetron deformation - could Authors explain what does that mean?
- Figure 9 - the picture of the single crystal should be of better quality, the sample is more important than the ruler
- Figure 10 - some of the captions on the picture blend with the picture itself
- lines 268, 274, 281, 311 - Tesla is not an SI unit
- lines 299-308 - inconsistent naming of the martensitic and austenitic structures with capital letters or not

In my opinion, one cannot 'translate' a single sample experiment into a fully operational sealing system. What is the source of the micro-texture morphology you have proposed in Figure 7? How can you control the steep wave-like texture of the Ni2MnGa surface? How well it would perform as a sealing, especially in temperatures higher than 35 deg C - additional cooling system for the cylinder sounds not that much practical.

The English is awkward, it seems that the text is composed using google translator. Although English is not my mother tongue I have to admit the text should be extensively edited, a help of a native speaker or a professional translator is strongly suggested.

Yours faithfully

Reviewer

Author Response

Dear Reviewer 1, 

     Thank you for your kindness and patience. We have finish the revisions. Please see the attachment.

Kind regards,

xiaolan

Reviewer 2 Report

In this work, the authors studied the variation of gap sealing technology of the hydraulic cylinder using the magnetic shape memory alloy. The research appears to be efficiently done and appropriately reported, however, the standard of English must be improved there are many errors that must be improved by a professional. Nevertheless, there are some questions and corrections that must be answered to improve and complete the document.

Abstract section: The abstract is a little bit confuse and missis some information like more results and conclusions, I suggest to authors follow these rules:

  1. One or two sentences on BACKGROUND
  2. Two or three sentences on METHODS
  3. Less than two sentences on RESULTS
  4. One sentence on CONCLUSIONS

Figure 2. Please, show in the Figure the deformation failure, you can use a circle in this region.

Lines 98-100. The authors compare the MSMA with GMM and PCM, indicating some advantages, however, they didn’t prove these statements with any reference. Please, indicate the references where these statements are written.

Table 1. Please, change the word “Billions” by the numerical value.

Lines 129-130. What is a "certain manufacturing accuracy and positioning accuracy"? Can you indicate the correct values of the dimensional and geometrical tolerances?

Line 163. Please the sentence “… the strain applied …” to “… the stress applied …”.

Line 266.  The authors wrote, “ … sample preload …”. Is the preload on compression?

Lines 269-271. Can the authors represent, schematically, the system to apply the preload, the magnetic field, the temperature control, and strain measurement?

Lines 274-275. They wrote that they used “strain gauges” to measure the “deformation rate” (I think it is “strain rate”). In which part of samples are located the strain gauges? How many strain gauges? Type and electrical (if they are resistive) characteristics of strain gauges?

Conclusions (lines 336—353). The must use a dot marker for each paragraph

Author Response

Dear Reviewer 2,

Thank you for your kindness and patience. We have finish the revisions. Please see the attachment.

Kind regards,

xiaolan

Round 2

Reviewer 1 Report

Dear Authors,

thank you for considering my comments. I will recommend the publication of your work.

Yours faithfuly

Reviewer